# The Use of Deep Brain Stimulation in Tourette Syndrome

**DOI:** 10.3390/brainsci6030035

**Published:** 2016-08-19

**Authors:** Ladan Akbarian-Tefaghi, Ludvic Zrinzo, Thomas Foltynie

**Affiliations:** 1Institute of Neurology, University College London (UCL), Queen Square, London WC1N 3BG, UK; ladan.akbarian-tefaghi.15@ucl.ac.uk; 2Sobell Department of Motor Neuroscience, University College London (UCL) Institute of Neurology, London WC1N 3BG, UK; l.zrinzo@ucl.ac.uk

**Keywords:** Tourette syndrome, TS deep brain stimulation, DBS

## Abstract

Tourette syndrome (TS) is a childhood neurobehavioural disorder, characterised by the presence of motor and vocal tics, typically starting in childhood but persisting in around 20% of patients into adulthood. In those patients who do not respond to pharmacological or behavioural therapy, deep brain stimulation (DBS) may be a suitable option for potential symptom improvement. This manuscript attempts to summarise the outcomes of DBS at different targets, explore the possible mechanisms of action of DBS in TS, as well as the potential of adaptive DBS. There will also be a focus on the future challenges faced in designing optimized trials.

## 1. Introduction

Tourette syndrome (TS) is a childhood neurobehavioural disorder affecting approximately 0.3%–0.8% of the paediatric population [1]. It is defined by the presence of at least one vocal and two motor tics starting before the age of 18, lasting longer than one year, with the exclusion of other causes [2]. Tics are often preceded by a premonitory urge [3] tend to occur daily in variable bouts and follow a waxing and waning course. The typical age of onset is five to seven years [4] with symptoms peaking at puberty and often remitting into early adulthood. In around 20% of patients, symptoms persist and have a detrimental effect on quality of life including social, professional and educational development [5]. Psychiatric co-morbidities are experienced in approximately 50%–90% of TS individuals, particularly obsessive compulsive disorder (OCD/OCB), attention deficit hyperactivity disorder (ADHD), deliberate self-injurious behaviours (SIBs) along with disturbances in mood and anxiety [6]. Current treatments include a combination of pharmacological and behavioural therapies. However, in patients with disabling refractory symptoms, surgical intervention, such as deep brain stimulation (DBS), may be considered a potential option for symptom improvement. 

DBS has been applied to patients with both hypokinetic and hyperkinetic movement disorders including Parkinson’s disease, dystonia and essential tremor, as well as neuropsychiatric disorders such as treatment-resistant depression and OCD [7]. Targets of DBS for TS are based on the postulated dysfunction of the basal ganglia-thalamo-cortical loops. A simplistic proposal is that aberrant activity in groups of striatal medium spiny neurons [8] lead to a decrease in the inhibitory output of the globus pallidus internus (GPi), resulting in disinhibition and the execution of involuntary cortical motor commands in the form of repetitive, stereotyped movements. A variety of methods, such as structural imaging in longitudinal studies of TS patients, have correlated smaller caudate nucleus volumes with severity of tics, as well as changes in the diffusivity of water molecules in the frontal lobe and thalamus as measured by diffusion tensor magnetic resonance imaging [9]. Decreased connectivity between caudate nucleus and lateral frontal cortex has been observed, supporting a cortical disinhibition theory for the disorder [10] though contradictory findings have also been reported [11] and the precise pathophysiology of TS remains unknown. 

The current manuscript attempts to provide an updated review of the published literature regarding the use of DBS for severe, medication refractory Tourette syndrome. A particular focus has been (1) to understand the potential mechanisms of action of this therapy based on neurophysiological recordings and neuroanatomical knowledge of the basal ganglia circuity; and (2) to make recommendations for the future evaluation of the use of DBS in TS.

## 2. Methods

A broad search was carried out using the databases Pubmed and OVID Medline with a variety of terms including “Tourette syndrome“, “GTS“, combined with “deep brain stimulation“ or “DBS“. Each target region for DBS, including the “thalamus“ “globus pallidus internus“ “anterior limb internal capsule“ “nucleus accumbens“, “globus pallidus externus“ were combined with the above terms. Only references in the English language were included. 

## 3. Literature Review

Currently, nine targets have been used in DBS for TS, including the thalamic centromedian parafascicular complex (CMPf), the cross point of the centromedian nucleus-substantia periventricularis-nucleus ventro-oralis nucleus (CM-Spv-Voi), the target of the nucleus ventro-oralis posterior-ventro oralis anterior-Voi complex (Vop-Voa-Voi), the globus pallidus internus (GPi) (anteromedial AND posteroventral regions), the nucleus accumbens (NA), the anterior limb of the internal capsule (ALIC), the subthalamic nucleus (STN) and the globus pallidus externus (GPe) [12]. Known connections between nuclei targeted by TS DBS, as well as other structures in the cortico-basal ganglia network are illustrated in Figure 1.

The rationale behind target choice has varied depending on whether tics are considered a movement disorder in which case sensorimotor areas such as the posteroventral pallidum have been stimulated or if they are considered to be a compulsion or a failure of inhibition wherein associative/limbic areas have been targeted [13]. The thalamus and GPi have been the most widely stimulated with a combination of targets used in some studies. A summary of the results of open label trials is presented in Table 1, and blinded trials in Table 2. The outcome measures in these studies relate to tics or co-morbidities and include the Yale global tic severity scale (YGTSS) [14] as well as the Modified Rush video rating scale (MRVRS) [15] which measure tic frequency, severity and impairment levels. The Yale Brown Obsessive Compulsive scale (Y-BOCS) [16] assesses the severity of OCD/OCB symptoms whilst measures of quality of life, anxiety and depression commonly include the Gilles de la Tourette quality of life (GTS-QOL) scale [17] State Trait Anxiety Inventory (STAI) and Beck depression inventory (BDI), respectively, which have been validated for use in TS.

### 3.1. Thalamic Targets

Motor patterns in TS are postulated to result in part from an increased thalamocortical drive due to inappropriate activation of striatal neurons [18]. There is an excitatory feedback loop from the thalamus to the striatum originating in the CMPf and midline thalamic nuclei [19], which are the most common thalamic targets for TS DBS.

The first report of DBS for the treatment of refractory TS by Visser-Vandewalle, used the same thalamic nuclei (centromedian-parafasciucular complex (CM/Pf), and ventral oral internus nuclei (Voi)) that Hassler and Dieckmann had targeted in stereotactic ablation in 1970. This individual was a 42 years old male who subsequently experienced a 90% improvement in his tics at 12 months [20] although using stimulation parameters with extremely high charge density, in comparison to those used in Parkinson’s disease. In 2003, the same group reported three patients including the above who showed a 72%–90% improvement in tics over a follow up of between eight months and five years [21]. In two of three patients the co-morbidities of SIB and OCD were also no longer present. The main adverse effects reported were either a reduction or increase in sexual drive in two patients along with reduced energy levels. In the larger series of 18 patients who were followed up between 3 and 18 months [22] YGTSS improvement varied between 24% and 79% in 15 of 18 patients with a concurrent improvement in co-morbidities. A longer follow up study [23] of 15 males after five to six years showed a mean YGTSS improvement of 73% and YBOCS of 42%. Adverse effects included a scalp erosion due to compulsive picking, and an abdominal haematoma. These and other open label studies [24,25,26,27,28,29,30,31,32,33,34,35,36,37,38] are outlined in Table 1 and strongly indicated that thalamic stimulation may have a beneficial therapeutic role in TS for both tic severity as well as comorbid symptoms.

However, the results of two randomised controlled trials showed a more variable response. In 2007, the anterior part of the CMPf complex was targeted in five male patients, with only a mean improvement in tics of 17% in the YGTSS when blindly comparing ON vs. OFF stimulation. However, three of the five responded with a 50% average reduction of tics following open label stimulation at three months, though no biomarker could distinguish responders from non-responders. Measures of OCD, depression and anxiety demonstrated a trend towards improvement, with a 44% increase in the YBOCS whilst one patient experienced a psychotic episode [39]. A more recent double blind crossover study targeted the CM-Spv-Voi assessing 6 male patients who were randomly assigned to three months on stimulation and three months off or vice versa. At a group level improvement in tic control during the off and on condition was 2.8% and 39.4% respectively. After one year open label follow up, these levels were maintained at 49.2% on YGTSS and 35.5% on the MRVRS [40]. 

### 3.2. Globus Pallidus Internus (GPi)

The GPi is considered to be functionally segregated into an anteromedial region which is part of the associative/limbic loop of the cortico-striato-thalamo-cortical circuit, while the posteroventral region is involved in the motor loop of the parallel circuit. However, there is much convergence of both limbic and motor inputs in both segments of the GPi [65]. Therefore, there has been much debate as to which sub-region might respond best to DBS for TS. The posteroventral target is familiar territory to functional neurosurgeons as it has been stimulated in other hyperkinetic movement disorders, such as Parkinson’s disease (PD) and various forms of dystonia.

#### 3.2.1. Posteroventral GPi

The results of open label studies of DBS of the posteroventral GPi region in TS patients [24,41,45,46,49,52,54,55,57,58] are also listed in Table 1. The range of tic improvement has been 20%–92.9% at 6–39 months follow up [41,45,46,52,54,55,57], with serious side effects including a small haemorrhage in the right GPi at surgery resulting in a persistent bradykinesia of the left hand in one 27 year old male [41] and complaints of depression [46]. In a four patient cohort [52], two patients were non-responders and two experienced an improvement in YGTSS of up to 88%, whilst in a long-term outcome study of 13 patients YGTSS improved by 52.1% with a wide range of 4.3%–83.6% over a mean follow up of 13–80 months) [58]. Stimulation in a 16 year old boy with TS and severe mental retardation led to lack of symptom improvement, attributed to his severe co-morbidity [49]. Despite the variable response at this target it may be preferred in patients with dystonic tics [66].

#### 3.2.2. Anteromedial GPi

The rationale for targeting the anteromedial GPi region is based on the associative-limbic connections that may impact the underlying urge driving the expression of motor and phonic tics. A summary of the results of open label trials is also presented in Table 1 [52,56,59,60,61] with the amGPi demonstrating tic improvement ranging from 38% to 71.5% [52,56,59,60,61] in patients followed up between 4 and 46 months, with anxiety reported as an adverse effect [56,59].

However, when evaluated using a randomised crossover trial methodology in 15 patients, comparison of blinded ON stimulation vs. OFF stimulation periods showed a mean YGTSS improvement of only 15.3% although with substantial variability between patients. With long-term follow up assessment, patients showed a 40.1% improvement in tics and 38.9% improvement in quality of life [64] leading to discussion about factors which may influence the outcome of DBS surgery when evaluated using blinded trial methods [67].

### 3.3. Other Targets

The Subthalamic nucleus (STN), External Globus pallidus (GPe), Anterior limb of Internal capsule (ALIC) and Nucleus Accumbens (NA) have been less frequent targets for stimulation [26,36,42,43,44,47,48,50,51,53,54,68]. The result of STN stimulation has only been reported in one patient who had both PD and TS with DBS resulting in a 97% improvement in tics along with relief of PD symptoms [51]. One male patient stimulated in the GPe region showed a 70.5% increase in YGTSS and a 75% improvement in anxiety [68]. Although limited by study design and sample size, the above results certainly suggest that the STN and GPe show promise as targets of stimulation and warrant further study.

The NA and ALIC have been used as a target in treatment-resistant OCD and have been chosen as potential stimulation targets in some circumstances owing to the frequent OCD comorbidity seen in TS [69]. Open label studies have shown tic improvement of between 25% and 45% [42,43,44,48,50] at up to 34 months follow up with adverse effects including rapid internal pulse generator depletion in one patient requiring 2 replacement in 36 months [48]. However, in one case report there was no benefit from NA stimulation [47].

### 3.4. Comparative Studies

In a randomised crossover study, three patients had electrodes placed in both the thalamus and the anteromedial GPi, stimulation resulting in a YGTSS improvement of 65%–96% for amGPi, 30%–64% for CMPf and 43%–76% in combination stimulation. Follow up for as long as 60 months in 1 patient demonstrated maintenance of tic improvement. Pallidal stimulation appeared to be more effective for tic reduction than thalamic or even combination stimulation. However, depressive mood, anxiety and impulsiveness were not affected by pallidal stimulation alone, whilst they showed improvement in thalamic or combination stimulation. The thalamic and pallidal targets gave rise to varying adverse effects with the former including arm paraesthesia and reduced libido whilst the latter included increased anxiety [62,63].

The ALIC, which has been used for sterotactic lesioning in OCD patients and contains fibers that travel between the orbitofrontal cortex and the anterodorsal thalamus was stimulated in a 37 year old female with TS. This resulted in a 25% improvement on the YGTSS, but caused apathy or hypomania as side effects. The target was subsequently switched to the CMPf of the thalamus [42,43] resulting in a YGTSS improvement of 32% with cessation of SIBs that had led to retinal detachment. In a four patient cohort, two patients who had previous surgery at the CM-Pf-Vo, subsequently had ALIC/NA stimulation as rescue surgery with combined stimulation demonstrating superior tic reduction (82.6% than ALIC/NA only (68.1%)) [43]. These comparative studies can inform whether specific targets have a differential effect on tics and co-morbid symptoms which could impact target selection for TS patients with different co-morbidities. 

### 3.5. Clinical Outcome and Targets

In a recent systematic review the thalamic targets of DBS were found to improve YGTSS scores by an average of 47.62% *p* < 0.001 with comparable results for GPi (55.32%), whilst the ALIC/NA regions showed a 44% improvement (*p* = 0.017) albeit in a smaller number of patients [70]. Direct comparisons between targets generally also indicate a lesser effect with ALIC stimulation [50,53]. The co-morbidity outcome across targets revealed a median 31.25% reduction in YBOCS and 38.89% in BDI, but without any statistically significant difference in the outcome of tics or co-morbidity. It should be considered that clinical outcomes are attributed to the target yet the anatomical brain regions ultimately stimulated by DBS depend on precise electrode placing, stimulation parameters, as well as individual anatomical variation [70]. The development of computer models using electrode location and extent of current spread has enabled investigation of the electric field distribution in individual patients using proton-density magnetic resonance imaging (MRI) scans. In one such study of five patients who had DBS targeting the amGPi, patient-specific stimulation confirmed this region to be the main stimulation field. However, three patients showed possible extension into GPe and internal capsule and in two patients with clear extension into the anterior GPe, there was a superior clinical effect. This highlights the importance of analysing the exact position of effective stimulation contacts and correlating outcome with patient specific stimulation fields in order to obtain more valid results about optimal target [71]. 

Although no definitive predictive factors for improved tic outcome have been identified, a positive correlation was observed between pre-operative YGTSS impairment score and tic reduction following amGPi DBS (*p* = 0.01), whilst the opposite effect was seen for pre-operative YGTSS tic severity and outcome following thalamic DBS. Furthermore there was no correlation between severity of co-morbidity symptoms or with stimulation parameters and tic outcome [70]. This meta-analysis did find a trend towards younger age and improved outcome, suggesting that the neurobiological underpinnings of TS are better impacted by stimulation in the younger population. Interpretation is limited by pooling of results from sub-regions of the thalamus or ventral striatum which could be improved by using individual co-ordinates and settings though these are not always available. These findings nevertheless remain informative and suggest that higher YGTSS impairment scores, tic severity, patient age and choice of preferred target should all be considered in the selection of patients for DBS. 

### 3.6. Inclusion and Exclusion Criteria

The suitability of DBS in TS patients depends on many factors yet there is much variability around specific inclusion and exclusion criteria across studies. Typically, core criteria focus on accurate diagnosis, high tic severity (typically a YGTSS score >35/50) and resistance to at least three different pharmacological agents [72]. Although psychiatric co-morbidity is a common component of TS, motor and vocal tics should be the main source of disability [7]. In addition, realistic expectations of DBS outcome and adequate social support [22], as well as compliance in attending appointments for outcome assessment or alteration of parameter settings is crucial when selecting patients for DBS. These should all be considered in a multidisciplinary team setting including a neurologist, psychiatrist, neurosurgeon and psychologist.

Exclusion criteria typically include major psychiatric disorders, pregnancy, current substance abuse or dependence and severe cognitive impairment, (supported by a study showing a lack of benefit in a 16 year old boy with TS and mental retardation undergoing DBS, although the outcome of this single case report should not be over-interpreted [49]). Other factors warranting exclusion include structural abnormalities on MRI, general contra-indications for surgery and patients with somatoform disorder [61] which have also been outlined in the European guidelines for TS [73]. Some centres have also used age as an exclusion criteria, restricting DBS to patients over 18 or 25 [73], based on the premise that TS may subside into adulthood and that there are ethical implications for placing patients under potentially unnecessary surgical risk. Nonetheless, adolescence is a crucial period for social, emotional and educational development and preventing younger patients with severe debilitating TS receiving treatment could impact negatively on independence and wellbeing into adult life. This is supported by beneficial reports in younger patients including a 17 year old with SIBs that prevented full time school attendance who showed markedly improved social integration post DBS [37]. Consequently, recent recommendations argue against strict age criteria and suggest that younger patients should be reviewed on a case by case basis, involving a local ethics committee.

### 3.7. Adverse Effects

Adverse effects can be classified into procedure or stimulation related events with some stimulation effects varying between targets. Stimulating the thalamic CM-Spv-Voi region has been associated with gaze disturbances or visual symptoms which are less often seen when targeting motor regions of the thalamus [34]. Other adverse effects include increased or decreased libido, which can result from modulating certain parts of intra-laminar thalamic nuclei involved in controlling sexual function [74]. In addition, arm paraesthesia, dysarthria and a case of psychosis have been reported. 

Stimulation of the pvGPi has been associated with increased anxiety, depression and memory impairment [33], whilst that of the amGPi has also been related to higher anxiety levels, dyskinetic limb movements and a case of hypomania [64]. Stimulation of the ALIC/NA was linked to side effects such as depression, hypomania, as well as a suicide attempt reported in an NA patient with known depressive episodes. The adverse effects of apathy, fatigue, dizziness and weight changes are more common across targets and in many instances adjustment of the stimulation parameters can diminish or eradicate these but target selection should aim to limit their effects. 

Procedure related adverse effects mostly centre around hardware malfunction and infections with the latter appearing to be higher in TS compared to other movement disorders. This was suggested by a retrospective study of 272 DBS patients, 39 of whom were treated for TS and showed an overall 3.7% infection rate whilst for TS this was 18 [75]. The higher rate of infection in TS patients was attributed to the compulsive picking at surgical scars in some patients [75] though altered immune function has also been postulated including lower T cell count [76], dysgammaglobulinemia, or an altered immunomodulatory effect of dopamine [77], though this requires further investigation.

### 3.8. Mechanisms of DBS Action in TS

DBS has demonstrated improvement in patient symptoms across targets yet much remains to be elucidated about its mechanism of action. Some studies have investigated the role of DBS in the dopaminergic modulation of striatothalamic pathways, supported by the postulated hyper-dopaminergic innervation in TS pathology and the use of dopamine antagonists in TS treatment. The first report of modulation of dopaminergic transmission with DBS was in a 22 years old patient who had undergone bilateral thalamic stimulation with a resultant improvement in tics [78]. Six months post surgery [^8^F] fallypride (FP) positron emission tomography (PET) scans, which quantified the striatal and extrastriatal dopamine 2/3 (D2/3) receptors, showed a 16.3% decrease in dopamine binding potential during on-stimulation compared to off-stimulation conditions, suggesting that DBS may indirectly cause a decrease in dopamine release. In addition, there was an increase in D2/3 receptor availability in the baseline on condition compared to healthy control subjects (*n* = 20) based on a published control group. A further study using FP PET scans on three patients receiving thalamic DBS demonstrated similar results in dopamine binding during DBS [79]. Although these studies are limited by small sample size and potential anaesthetic effect, they indicate that dopamine modulation may be a component of the therapeutic impact of DBS.

It had previously been shown using electroencephalography that premotor potentials did not precede simple motor tics in TS suggesting that the cortex was not their site of origin [80,81] though more recent contradictory findings have also been published [82]. Microelectrode recordings in thalamic nuclei during DBS surgery has typically shown a burst-firing pattern with an interburst interval ranging from 0.12 to 0.4 s within the low frequency (delta/theta) range (2.5–8 Hz) [83]. Similar results were obtained in a study that demonstrated oscillatory activity at low frequencies [84], which has also been observed in other hyperkinetic movement disorders, such as dystonia [85] suggesting that low-frequency activity and decreased thalamic beta activity could contribute to the pathophysiology of TS. This is supported by correlation with clinical phenotype in a patient who had few tics but severe OCD showing significantly fewer low-frequency oscillations than in local field potentials (LFPs) recorded in patients with severe tics. Further, increased gamma activity (25–45 Hz) recorded from the CM in awake patients with TS (*n* = 5) correlated with tic improvement and demonstrated that DBS increases the power of LFP gamma oscillations [86]. Therefore, shifting basal ganglia thalamic oscillation power from low to high frequency may be one of the effects of DBS in TS patients, supported by the correlation between increased CM gamma activity and clinical tic reduction [86]. Short-term neuronal mechanisms of DBS in TS have also been investigated using primate models of basal ganglia-mediated motor tics [87]. High frequency stimulation in the anteromedial GPi was shown to significantly reduce the amplitude of tic related phasic changes in the pallidum most likely caused by cellular activity temporally locking with the stimulation pulse [87].

Neuroplasticity is also being recognised as relevant to the effects of DBS supported by the reported shift in magnitude of long-latency response components across stimulation blocks [87]. However in a study investigating thalamic LFP recordings in a 48 years old male with Tourette syndrome 12 months after DBS, LFPs remained in the low frequency range as observed a few days after initial DBS implantation, suggesting that DBS may not cause a persistent change in LFP oscillation pattern [88] though this requires investigation in more patients. Evidence has implicated microglia in the modification of synaptic connections and plasticity which could contribute to functionally immature control networks typical of that seen in TS. The role of DBS in altering the immunobiology of the brain in TS is an interesting avenue of research [89].

Mechanisms of DBS in TS should continue to be explored as they inform both the pathophysiology of the disorder and ways in which DBS delivery can be optimized in patients. When considering the search for an optimal target, the high interconnectivity of basal ganglia structures means it is likely that stimulation in one area will have profound downstream or upstream effects in others and rather than the target being a specific focus of pathology, it may instead block propagation of aberrant signals through a local network [87]. As DBS currents spread through anatomical connections, it may be useful to consider the effect of target stimulation on these different brain networks through a variety of imaging techniques. Functional magnetic resonance imaging (fMRI) compatible DBS systems could be used to map cortical-subcortical circuits [90] and patient specific tractography activation models (TAMs) could identify white matter pathways and monitor the projections being activated by DBS in individual patients [91]. Resting state functional connectivity MRI, which has previously been used to identify thalamic DBS targets based on connectivity to brain regions in tremor could be applied in a similar way to tics in TS [92]. These methods may help motivate a shift to the stimulation of specific brain networks and potentially customise treatment for varying patient phenotypes. 

### 3.9. Adaptive DBS in TS

Further research into neurophysiological recordings in TS patients can also inform the potential future application of adaptive rather than continuous DBS which may be particularly relevant due to the paroxysmal nature of TS [93]. An adaptive DBS system would allow the pathological neural activity of patients to be used as feedback through variables, such as LFPs recorded by DBS electrodes, informing when tics will arise and leading to a responsive alteration in stimulation to suppress their onset. 

The finding that increased gamma band activity and reduced alpha band power in the CM complex of the thalamus correlated with superior clinical outcome [86] could be the starting point for developing specific neural markers correlating with tic onset. Moreover, the GPi has been seen to exhibit low frequency oscillations that precede the electromyographic recording of tics by 50 ms or more, and may mark the premonitory urge of motor tics [88], whilst LFP recordings in ALIC/NA, have shown high beta power oscillations, a potential physiological indicator of OCD activity [88]. However, the reliability of feedback algorithms and the accuracy of correlation of variables such as LFPs with clinical symptoms requires much investigation, particularly as the latter can differ between tic types and individual patients [94] A proof of principle study showed the beneficial effects of scheduled DBS in TS patients [32] and though a closed loop system was not employed it may provide the foundation for applying neuromodulatory approaches. The benefits of an adaptive DBS system pertain to the reduced energy expenditure leading to a prolonged battery life [95], as well as a reduction in stimulation related side effects, such as fatigue or anxiety, potentially further improving patient quality of life. 

Nonetheless, the use of adaptive DBS remains highly experimental in all conditions, not least until a consistent neurophysiological biomarker for tic urge is identified and issues of sustained efficacy and potential side effects have been thoroughly investigated in future trials and studies.

## 4. Discussion

Stimulation applied to different cortico-basal ganglia thalamic network structures appears to improve motor and non-motor symptoms in TS. However, over-interpretation of ‘between target comparisons’ of these results is unwise as most data are derived from case reports and small prospective series with wide variability in methodology. Although more double-blind controlled studies are required, designing these to ensure adequate optimization of stimulation settings as well as maintenance of patient and investigator blinding can be a challenge [67].

Therefore, study design and patient selection should be carefully considered in the future in order to investigate the relative strength of treatments across studies. 

Establishing an optimal target has been controversial as the precise pathophysiology of TS is not yet elucidated and stimulated structures have high interconnectivity. To date, the most commonly used targets have been the thalamus and GPi, which have shown variable but overall promising results. Many patients with electrodes targeting the GPi will have electrode trajectories that also straddle the GPe. Reviewing the data in Table 1 and Table 2, the optimal stimulation parameters often include the more dorsal electrodes therefore potentially also preferentially delivering stimulation to the GPe fibres. 

Only single studies have investigated the STN and the GPe despite demonstrating beneficial effects in tic reduction. Further, the STN is known to have an excitatory influence on the GPi, which could be manipulated to affect both the limbic and sensorimotor circuits to a greater extent than using the GPi or thalamus alone. Further insight into the cortico-subcortical networks stimulated by DBS through different imaging modalities will further inform optimal targeting. 

Looking closer at patient specific factors related to variability of response could help reveal predictive factors for improved outcome. Results of a meta-analysis suggesting that higher pre-operative YGTSS impairment and younger age in patients targeted in the GPi region, correlated with better tic outcome in the may also inform future inclusion and exclusion criteria. Furthermore, correlating clinical outcome to specific fields of stimulation based on patient specific computer models, may contribute to a more accurate mapping of the optimal target within structures like the GPi/GPe. 

Study outcomes have focused on tic reduction using rating scales, such as the YGTSS, and although these are validated and include measures of overall impairment, they are not always an accurate reflection of tic impact on self-esteem and socio-professional life. The social integration of TS patients should be a considered a vital aim of the DBS procedure, determined not only by a reduction in tics, but often more importantly, by that of co-morbid symptoms. Consequently, it is crucial that studies uniformly use scales such as the GTS-QOL, STAI and BDI in order to compare quality of life and co-morbidity outcomes across targets. Further, supportive accounts from caregivers can also increase the accuracy of clinical evaluation. 

In order to further assess the efficacy of thalamic DBS there is a need for larger, multi-centre trials with careful consideration of optimal trial design to ensure the outcome of this intervention can be objectively assessed. One proposal would be to perform a direct comparison of thalamic (CMPf) DBS against anteromedial GPI/GPe DBS with the latter electrodes deliberately straddling both GPi and GPe targets. Intraoperative and post-operative recordings from externalized electrodes will continue to be informative regarding the neurophysiological changes associated with tics and OCD/OCB. Confirmation of dopaminergic receptor occupancy post DBS using functional imaging techniques would be useful. 

To compare the clinical efficacy of two targets requires a sufficiently long period of time for resolution of surgical swelling and wound healing and stimulation parameters can require long periods before considered optimized. Objective confirmation of efficacy must also include an off stimulation assessment, brief if necessary to allow the impact of stimulation to be distinguished from placebo effects. There is now a worldwide consortium of clinicians including functional neurosurgeons interested in DBS and the design of such future trials are the subject of ongoing discussions. 

## 5. Conclusions

Despite variable outcomes between patients, DBS for TS has shown much promise across targets. The careful design of randomised trials, use of comparative studies and imaging modalities in DBS systems can inform target selection. Considering that the ultimate aim of this procedure is the social integration of patients, future studies should continue to address the treatment of both tics and co-morbidities. Further elucidating the mechanisms of DBS action can help enable its optimization, inform the pathophysiology of TS and future potential applications of adaptive DBS. 

## Figures and Tables

**Figure 1 brainsci-06-00035-f001:**
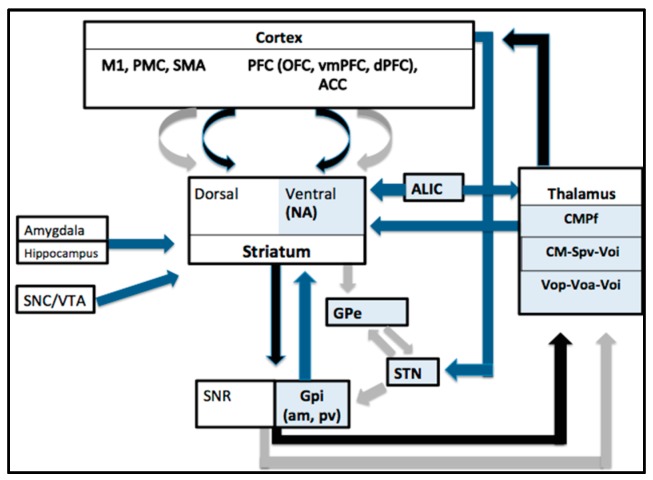
Simplified schematic showing the main connections of the cortico-thalamo-cortical network, with the nuclei targeted in TS DBS in blue boxes (NA = nucleus accumbens, ALIC = anterior limb internal capsule, CMPf centromedian parafasciular complex, CM Sp Voi, centromedian nucleus-substantia periventricularis-nucleus ventro-oralis nucleus, the nucleus ventro-oralis posterior-ventro oralis anterior ventro-oralis complex (Vop-Voa-Voi), GPe = globus pallidus externus, STN = subthalamic nucleus, GPi (am, pv = anteromedial, posteroventral globus pallidus internus). The direct pathway is shown with black arrows, the indirect pathway with grey arrows and other connections shown with blue arrows. Projections from the primary motor cortex (M1), the pre-motor cortex (PMC) and the supplementary motor area (SMA) are predominantly directed to the dorsal striatum (putamen and caudate) whereas fibers from the prefrontal cortex (PFC) including the orbitofrontal cortex (OFC), ventromedial PFC (vmPFC) and the dorsal PFC (dPFC) as well as the anterior cingulate cortex (ACC) mostly project to the ventral striatum (nucleus accumbens and rostroventral most aspects of caudate and putamen). The main output nuclei are the GPi and the substantia nigra pars reticulata (SNR). Other regions, such as the substantia nigra pars compacta (SNC), also have connections with the striatum.

**Table 1 brainsci-06-00035-t001:** Summary of open label trials.

Study	Target	Sample Size, Sex (Age, Years)	Follow up (Months)	Stimulation Parameters	Tic outcome (Improvement in YGTSS or MRVRS)	Comoribidity Outcome	Adverse Effects/Comments
Vandewalle et al. (1999) [20], Visser-Vandewalle (2003) [21]	CMPf, SPv Voi	Three males (42, 28, 45)	60, 12, 8	Bipolar all contacts 100 Hz, 210 µs, 2.4 V right, 2.2 V left; Right double monopolar 65 Hz, 210 µs, 3 V; left monopolar 100 Hz, 210 µs; Right monopolar 2.8 V, 130 Hz, 210 µs; left one monopolar 2.4 V, 100 Hz 200 µs	90%, 72%, 83%	OCD and SIB disappeared in all 3.	A slight sedative effect in all three; 2 had increased or decreased libido.
Diederich et al. (2005) [41]	pvGPi	One male (27)	14	Monopolar 2 V, 185 Hz, 60 µs	73% (previous YGTSS 83)	No apparent change in the mild compulsions of the patient. Significant reduction of anxiety/depression.	Small symptomatic haematoma right pallidum bradykinesia of left extremities
Flaherty et al. (2005) [42], Shields et al (2008) [43]	Anterior capsule CMPf, SPv Voi	One female (37), same patient operated at different target	18, 3	Bipolar 4.1 V, 185 Hz, 210 µs, 7 V, 90 µs, 185 Hz	YGTSS 25%, 32%	Self-injury stopped (previous retinal detachment led to blindness in one eye), but vision stabilized post DBS	Electrode breakage therefore same patient had stimulation at different target. The AC site caused altered mood and impulse control disturbance.
Ackermans et al. 2006 [24]	CMPf, Voi pvGpi	One male (45), one male (27)	12	Monopolar 6.4 V, 130 Hz, 120 µs bilaterally; Monopolar 3.1 V, 170 Hz, 210 µs bilaterally	Tics 20 to 3/min, Tics 28 to 2/min	Obssessions and compulsions (measured by the revised Padua inventory) improved by 62% in the ON than OFF condition	CMPf patient is the same as in Vandewalle study- he experienced vertical gaze palsy and decreased libido. Both patients complained of reduced energy.
Kuhn et al. 2007 [44]	NAC, internal capsule	One male (26)	30	Tetra monopolar 90 µs, 130 Hz, 7 V	41% with YGTSS, 50% MRVRS (Previous YGTSS 90 and MRVRS 18)	Reduction in SIB, reduction in OCD by 52% based on Y-BOCS	No significant adverse effects
Bajwa et al. 2007 [25]	CMPf, SPv Voi	One male (48)	24	Bipolar 2 V, 130 Hz and 90 µs	66% (Previous YGTSS score 35.5 and overall 83)	YBOCS improved 75%	Approximately 14 programming sessions required over 2 years. No serious adverse effects.
Shahed et al. 2007 [45]	pvGPi	One male (16)	6	Monopolar 5 V, 160 Hz right, 145 Hz left, 90 µs	84% (YGTSS)	YBOCS improved 69% (only obsessions not compulsions).	No surgical adverse effects
Dehning et al. 2008 [46]	pvGPi	One female (44)	12	Monopolar 4.2 V, 145 Hz, 210 µs	YGTSS 88%	Patient did have SIB including self-biting and beating but outcome not reported.	Complaints of depression. Vertigo and stomach aches in first few months. No serious adverse effects.
Zabek et al. 2008 [47]	Right NAC	One male (31)	28	Not reported	80% (15-minute videotaped exams)	None reported	Unilateral right side only.
Neuner et al. 2009 [48]	NAC	One male (38)	36	Double monopolar 6 V, 145 Hz, 90 µs	YGTSS 44% mRVTRS 58%	YBOCS 56%	Rapid IPG depletion-2 replacements in 36 months. Incidental finding was that patient no longer wanted to smoke.
Dueck 2009 [49]	pvGPi	One male (16)	12	Monopolar 4 V, 130 Hz, 120 µs	No improvement	Co-morbidity was severe mental retardation which was not affected	Not reported
Servello et al. 2008 [22]	CMPf Voi	15 males, 3 females (17–47)	3-18	Bipolar 2.5–4 V, 90–120 µs, 130 Hz	YGTSS 65%	None reported	Transient stimulation induced vertigo, poor scalp incision healing due to repetitive touching requiring body shield.
Servello et al. 2009 [50]	CMPF + ALIC/NA in 3, 1 had only ALIC/NA	Three males, one female (25, 31 ,37, 47)	10–26	3 monopolar, 1 bipolar 4–4.5 V, 130–160 Hz, 150–180 µs	Variable, mostly slight improvement in tics	Sight improvement in OCD	None reported
Servello et al. 2010 [26]	Total 79 procedures 36 patients) CMPf/Voi (67) pvGPi (2)ALIC-NA (10)	25 males, 6 females (17–57). 4 additional patients received leads in multiple targets and one in ALIC/NA	3–48	At last follow up 2–5 V, 90–140 µs, 60–180 Hz	YGTSS 47% mean improvement	17% mean improvement in YBOCS	2 patients had stimulator switched off, reporting unsatisfactory results, 2 had surgical revision of pulse generator due to infection, and one had hardware failure.
Martinez-Torres et al. 2009 [51]	STN	One male (38)	12	Monopolar 3 and 3.2 V, 130 Hz, 60 µs	76%	Not reported	Patient also had Parkinson’s disease which was the indication for DBS in STN
Martinez-Fernandez et al. 2011 [52]	pvGPi (2), amGPi (2), pvGPi (1) (then changed to anteromedial region after 18 months)	4 males (21–60), 1 female (35)	3–24	PL GPi: Monopolar 2.5 V, 150 µs, 170 Hz, 2.5 V, 60 µs, 130 Hz; AM GPi: Monopolar 4.2 V, 60 µs, 160 Hz, Monopolar 4 V, 210 µs, 130 Hz; PL then AM GPi: Bipolar 3.6 V, 60 µs, 20 Hz for both	Anteromedial group: MRVRS 54% , Posterolateral group (3 patients, with one experiencing worsening symptoms and responders average 37% improvement). YGTSS 38% improvement versus 20% for amGPi and pvGPi.	YBOCS mean change of 26% at last follow up	Complaints of agitation in 1, anxiety in 2, weight gain in 1, infection in 1 requiring repeat removal of battery.
Idris et al. 2010 [27]	CMPf, Voa	1 male (24)	2	3.5 V, 120 µs, 130 Hz	No scales reported, but tics noted to improve	Not reported	Postoperative bilateral subcortical haematomas attributed to low factor XIIIA
Ackermans et al. 2010 [28]	CMPf, SPv, Voi	2 males (42,45)	72–120	Amplitude L1, 8 R1, 5 130 Hz, 90 µs Amplitude L8, 5 R8, 5 100 Hz 150 µs At long-term follow up	Patient 1 (5 years) tic improvement 90.1%, maintained at 10 years (92.6%). In patient 2, after 8 months 82% slightly decreased at 6 years (78%). Video tic rating scale used measuring vocal and motor tics/10minutes.	Not reported, but in one patient “psychopathology“ reported to remain. Compulsions said to have disappeared in both patients. In one patient with depression, only slight decrease after surgery.	Both patients reported reduced energy. Both experienced traction of the lead in neck. One patient experienced a decrease in erectile function whilst the other had increased sexual drive. Both reported some visual blurring. One patient reported decreased verbal fluency and learning.
Burdick et al. 2010 [53]	ALIC/NA	One male (33)	30	Not reported	YGTSS 15% reduction at 6 months and thereafter until 30 months of last follow up remained unchanged. Initially had mild tics with a pre-operative MRVRS score of 5.	Not reported	No reported adverse effects.
Kaido et al. 2011 [19]	CMPf	One male (20) ,two females (19, 21)	14–21	Tetrapolar bilateral 2.3 V, 210 µs, 130 Hz; Tetrapolar Left 2.1 V, right 2.3 V, left 210 µs, right 180 µs, 130 Hz; bilaterally Tripolar Right 2.6 V, left 2.5 V, 180 µs, 80 Hz	Tics (52%–71%) Social impairment (56%–71%)	Not reported	No reported side effects.
Kuhn et al. 2011 [29]	Vop-Voa-Voi (unilateral)	One male (39), one female (27)	12	Bipolar 4.5 V, 130 Hz, 120 µs; Bipolar 3.1 V, 90 Hz, 120 µs	YGTSS 75%–100% MRVRS 77%–100%	BDI-no negative impact	Reduced verbal fluency at one year in both patients.
Dehning et al. 2011 [54] (one patient previously reported in Dehning 2008 [46])	pvGPi	Three females (25–44), one male (38)	5–13	4.2 V, 4 V, 3.8 V; 210 µs, 150 µs, 150 µs, 145 Hz, 130 Hz, 130 Hz (3 females) 3.5 V, 180 Ms, 130 Hz	2 patients responders (64% and 88% improvement YGTSS) 2 patients non-responders 1 female 2 male	Not reported	Lead revision was done in a non-responder without improvement.
Dong et al. 2012 [55]	pvGpi (unilateral R)	Two males (41, 22)	12	Quadripolar, 3.5 V, 2.8 V, 90 µs, 160, 130 Hz	YGTSS 53.1%–58.5%	None reported	No apparent adverse effects.
Cannon et al. 2012 [56]	amGPi	8 males (22–50), 3 females (18–34)	4–30	Quadripolar, initial stimulation 1 V, 60 µs, 130 Hz, adjusted at follow ups	YGTSS motor 48% vocal 56.5%	Mean YBOCS reduction 59%, HDRS 74%, GTS-QLS 102%	Complications from hardware malfunctions in 3 (due to SIB, MVA, and unknown). 1 patient did not tolerate DBS and switched it off. Anxiety in 2 patients
Duits et al. 2012 [30]	CM-Spv-Voi	One male (21)	23	Not reported	YGTSS- worse with DBS Pre-op-42 Stim OFF-12 Stim on-39	Y-BOCS Pre-op: 20 Stim OFF-8 Stim ON-7	Severe post-operative complications including psychogenic paroxysmal hypertonia.The patient may have had a somatoform disorder, that may contra-indicate DBS.
Savica et al. 2012 [31]	CMPf	2 males, one female (17, 17, 35)	1	17 years old males: Bipolar 3.7 V, 120 µs, 117 Hz; Monopolar 2.5 V, 90 µs, 130 Hz. 35 years old female: Bipolar 4.1 V, 120 µs, 107 Hz	YGTSS 70% mean improvement	No formal assessment, but co-morbid symptoms appeared stable or slightly improved.	Adverse effects related to stimulation such as mild paraesthesias but corrected with programming changes.
Okun et al. 2013 [32]	CM (scheduled stimulation)	3 females, 2 males (28–39)	6	Not reported	YGTSS 19% mean improvement mRVRS 36% mean improvement	YBOCS, HDRS, QOLAS did not improve.	No serious side effects.
Porta et al. 2012 [23]	Cm-Pfc-Voa	15 male, 3 females (17–47)	5–6 years	2.5–4 V, 60–120 µs, 130 Hz	YGTSS 73%	YBOCS 42% (*p* = 0.003), STAI-46%, BDI−55%	One patient developed poor healing of scalp scar due to compulsion to touch, the other developed abdominal haematoma where pulse generator was located. Majority had some minor side effects if voltage> 4 such as vertigo or blurring of vision.
Motlagh et al. 2013 [33]	Midline thalamic	4 males (16–44)	6–95	0.1–5 V, 60–120 µs, 60–200 Hz (4 bipolar one tripolar)	YGTSS- Greater improvement in the 2 younger patients (67%–85%), compared to older patients (7%–20%).	YBOCS-100% improvement in on patient but minimal change or worsening in others, HDRS and HARS-no change	44 year old male picked compulsively at chest and cranial incisions so DBS was removed due to infection. 42 year old male had DBS system removed due to lack of therapeutic effect.
Dong et al. 2014 [57]	pvGpi	1 male (33)	39	Monopolar 2.8 V, 90 µs, 130 Hz (frequency then reduced to 65 Hz 33 months)	YGTSS 92.9% at 33 and 39 months	YBOCS went from 18 to 0HAS and HDS markedly reduced	Supports that low frequency stimulation may be an optional therapeutic strategy in some patients
Zhang et al. 2014 [58]	pvGpi	12 males (16–34), 1 female (21)	13–80	Not reported	YGTSS Mean 52.1% (13–80 months).	GTS-QOL improved by a mean of 45.7% (range, 11.0%–77.2%).	Not reported
Sachdev et al. 2014 [59]	amGPi	14 males, 3 females (17–51)	8–46	Mean at follow up: 4.14 V, 95.2 µs, 139.4 Hz	Overall 48.3% reduction motor tics, 41.3% in phonic, 1 month 70.6% of patients had >50% reduction in YGTSS	YBOCS average 62% improvement (*p* = 0.001), 39% improvement in GTS-QOL (*p* < 0.001)	Lead breakage in 4 patients, one patient had infection, 2 transient anxiety, 1 dizziness, 1 poor balance.
Huasen et al. 2014 [60]	amGPi	One female (19)	12	Bipolar 2.9 V, 180 Hz, 180 µs	YGTSS 55% improvement	Not formally assessed	Patient had a cervical cord contusion secondary to violent cervical tics, with DBS used to preserve limb function and led to improvement in her neck extension tics
Zekaj et al. 2015 [38]	CMPf, Vo	1 male (17)	84	Not reported	At 12 months YGTSS 58.2%	Not reported	5 years post surgery DBS removed, and patient had stabilized despite stimulation. 2 year later continues stable. Symptoms may have resolved spontaneously. However, supports DBS in younger patients.
Huys et al. 2016 [34]	Ventro- anterior and ventrolateral motor parts of thalamus	Three female, five male (19–56)	12	1.3–3.7 V, 80–130 Hz, 60–150 µs	YGTSS and mRVRS at last follow up average 58%	In 5 no OCD at baseline, 2 mild and 1 severe. No significant effect on OCD comorbidity. Significant effect on quality of life improvement.	One patient had mild infection of subcutaneous pulse generator one patient had disturbance of eye motility, tremor of lower jaw. 1 patient had suicidal thoughts, 1 dysarthria.
Smeets et al. 2016 [61]	Anterior GPi	Three males and two females (35–57)	5–38	Tetrapolar 2.2–5.6 V, 180-360 µs, 100–180 Hz	YGTSS 71.5% average	In the three patients with baseline OCB (12%, 35% and 100%).	Two males had previous CMPf stimulation but due to side effects such as gaze disturbance, switched to GPi after 6 and 8 years after infection with IPG replacement. Other adverse effects included apathy in 2 patients, weight loss and agitation in 1 patient
Cury et al. 2016 [35]	CMPf	One male (23)	18	Not reported	YGTSS 70.5% Subscore impairment 60%	Patient did not have OCD 53% improvement on hospital anxiety scale (HAS)	None reported
Servello et al. 2016 [36], Testini et al. 2016 [37]	Vo-Cm-Pf-30, Vo-Cm-Pf and NA-ALIC-2, NA-ALIC (in 3 patients as single target, 3 as rescue therapy) amGPi (1 patient and 2 as rescue) CMPf	14 females, 34 males (total 48), 37 included in final analysis (17–57) 18 males, 3 females (17–46)	Up to 4 years 2–91	130 Hz initally in all patients, 60–120 µs, 2.5–4.5 V Bilateral quadripolar, specific parameters not reported	For remaining 37 patients Mean postoperative decrease in YGTSS 63% Reduction of more than 50% seen in 78.4% patients Average YGTSS 54% (46% motor, 52% vocal and 59% impairment). All but two patients reported marked reduction in tic severity and quality of life	35 OCD, 25 both OCD and depressive disorder. In 4 patients with moderate/severe depression, BDI improved by 45%, in the other 2 only slightly or inconsistently. All patients had psychiatric co-morbidities but outcome not reported.	In 11 patients the device was removed due to inflammatory complications or poor compliance and these were not included in final analysis. NA-ALIC was the single target in 3 patients, joint target in another 3, and rescue for further 3 patients. PvGPi targeted in one patient who then required rescue surgery in NA-ALIC. 12 patients had skin erosions requiring surgical intervention. 1 patient underwent wound revision due to scalp erosion and wound infection. Three years before, one patient had pallidal DBS with no apparent benefit. Postsurgical adverse effects reported on neuropsychological evaluation included occipital headache, and memory loss (including temporary anterograde amnesia)

**Table 2 brainsci-06-00035-t002:** Summary of randomised trials.

Study	Target	Sample Size, Sex (Age, Years)	Follow up	Stimulation Parameters	Effect on Tics Severity/YGTSS or MRVRS	Effect on Comorbidity	Adverse Effects/Comments
Maciunas et al. 2007 [39]	CMPf, Voi	Five males (18–34)	3	Variable polarity 3.5–3.6 V, 90–210 µs,130–180 Hz	Double blind comparison during first 4 weeks showed a 17% improvement. At 3 months 44% (mean) Non-responders with 4.3%–260% tic exacerbation	Mean score improvements: YBOCS 44%, BDI-2 60%, Hamilton anxiety scale, (HAS) 51%	One patient experienced acute psychosis during randomised period which was successfully treated. Overall three responders and two non-responders
Houeto et al. 2005 [62], Welter et al. 2008 [63]	CMPF and anteromedial GPi	Two females, one male (36, 30, 30)	60, 27, 20	CMPf: double monopolar 1.5–1.7 V, 60 µs, 130 Hz, Gpi: Single or double monopolar 1.5–3.5 V, 60 µs, 130 Hz	YGTSS cross-over period, (a)-AmGPi: (65%, 96%, 74%, (b)-CMPf: 30%, 40%, 64%, (c)-Gpi and CMPF 43%, 60%, 76% (after 60 months)	One patient previously had major depressive disorder and self-injurious behaviours and impulsiveness. Depressive mood, anxiety and impulsiveness tended to decrease with thalamic and pallidal stimulation but not paillidal sitmulation alone. None of the patients had OCD.	Reduced libido in one patient having thalamic stimulation. Lethargy, anxiety reported under pallidal stimulation and vertigo under higher intensity stimulation; arm paraesthesia under thalamic. Pallidal better than thalamic stimulation and both better than sham stimulation.
Ackermans et al. 2011 [40]	CM-Spv-Voi	6 males (completed full trial) (35–48 years)	36	1.3–7 V, 60–210 µs, 70–130 Hz. Monopolar stimulation in three patients and bipolar stimulation in the other three patients.	YGTSS at blinded ON compared to OFF stimulation was significantly lower (37%). After one year 49% and MRVRS 35%.	No significant difference found between the behavioural disorders and mood in the ON and OFF stimulation conditions.	One small haemorrhage ventral to electrode, one infection of pulse generator, subjective gaze disturbances which resolved after 6 months; all patient reported reduced energy levels. All patients when further questioned had subtle changes in oculomotor function, from visual disturbance to blurred vision and fixation problems, with no objective abnormalities detected with investigations. One younger patient with very severe SIB and life-threatening tics developed hypertonia, mutism and repeated fainting which needed extensive diagnostic evaluation- she was not randomised and considered loss to follow up. Only two patients completed the full 3 months on and of stimulation periods.
Kefalopoulou et al. 2015 [64]	AmGPi (13 patients), pvGPi (2 patients due to dystonic features).	11 males 4 females (25–55 years), 14 randomly assigned 13 completed assessments in both blinded periods. All 15 received stimulation in open-label phase	20–60	In blinded phase, 9 patients had monopolar and double monopolar in 4	YGTSS from off to on stimulation during blinded crossover period was 15.3%. Increased to 40.1% in open label phase	YBOCS-modest non-significant improvements Significant improvement in BDI GTS-QOL 38.9%	2 patients experienced infection of hardware requiring removal of leads and pulse generator with antibiotic treatment who were re-implanted 22 months later. One patient experienced deterioration of tics and hypomanic behaviour during on-stimulation periods, requiring stimulation parameter alterations and benzodiazepine treatment. 23 non-serious adverse events occurred, 15 of which resolved. During blinded phase, 6 patients had no clear benefit (<10% improvement in YGTSS) but most of these had more significant improvement in open label phase when parameters could be optimized. In open-label phase, 4 patients had less than 20% improvement n YGTSS.

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
