# Peer review of "The Use of Deep Brain Stimulation in Tourette Syndrome"

_brainsci, 2016, doi:10.3390/brainsci6030035_

Round 1

Reviewer 1 Report

This article, a comprehensive review of the literature, of deep brain stimulation (DBS) in Tourette syndrome (TS), focusing on target selection and future considerations.

Line 21:  Should reference the definition of TS (DSM-5)

Line 26: “severe” is redundant

Line 29: Cited source (reference 5) does not appear to be correct as it is of a single case report.

Line 29: Consider defining self-injurious behaviors (to differentiate these deliberate actions from tics that may cause injury).

Line 35/37: Consider citing original sources for these statements.

Line 41: What are diffusivity changes?

Line 49: More recent studies actually show evidence of increased, rather than decreased connectivity, in TS particularly between the insula and frontostriatal areas (Tinas et al. Mov Disord, 2015).

Line 57: Author state that 9 targets have been used but only 8 are listed.

Line 149: It should be mentioned that PV GPi is also a target in Parkinson’s disease

Line 154: Consider clarifying the “serious side effect” of hemorrhage may be a function of the larger number of patients with this target.

Line 164:  What are the percentages (41.3-71.5%) measuring; why is this separate from the 13 patients described on the next line with YGTSS score % improvements?

Line 182: Discussion of more dorsal electrodes corresponding to GPe stimulation should be reserved for discussion, unless specifically mentioned in those studies, in which case citations should be provided.

Line 211: What is meant by TS phenotypes?  Rather do you mean target selection based on comorbid conditions?

Line 214: Why are Thalamic and GPi targets lumped together as more effective as compared to ALIC /NA when estimates of improvement are more closely related between Thalamic and ALIC/NA targets than GPi?

Line 227: The phrase “generating more information” is ambiguous.

Line 231: Anteromedial GPi should be specified.

Line 253: I would advise caution in readily linking the patient’s mental retardation as a reason for poor response of TS as it was merely hypothesized in a single case report.

Line 285-287: A citation should be provided.

Line 305: Although the authors staste that “premotor potentials did not precede simple motor tics in TS”, some studies did show bereitschaftspotential prior to tics (van der Salm et al, JNNP 2012;83:1162-7).  The authors should comment on whether these cortical findings or the findings from thalamic recordings correlate with premonitory urges that often preced motor tics.  What about the relationship between LFPs and premonitory phenomenon?

Line 321-322: A citation should be provided.

Line 347: Caution must be taken in discussion of this section as statements made about brief/episodic use of DBS and its impact on symptoms/side effects may not be well supported.

Line 355: What is the relevance of persistent physiological findings (alpha/gamma band activity) when looking for episodic changes for an adaptive/reactive system?

Line 380-381: As you cite 1 case for each, “few studies” may be a misleading overestimation.

Line 386: The authors state that “Results of a meta-analysis suggesting that higher pre operative YGTSS impairment and younger age correlated with better tic outcome”, but they noted earlier that this was true only with GPi as a target.

Reviewer 2 Report

Due to the complexity of the syndrome  and the differents phenotype of patients so many differents targets have been used so far in DBS. I think that the goal of this procedure in Tourette ought to be social integration of these patients. In most cases social impairment is due to behavioural comorbidities rather then tics. This consideration must be stressed. Maybe we must prefer to choose a "limbic" target instead of a "motor" one but obviously we need a larger number of treated patients to verify this conclusion.

Round 2

Reviewer 1 Report

I believe the authors adequately addressed the reviewiers' comments.